# Individual Differences in the Vocal Communication of Malayan Tapirs (*Tapirus indicus*) Considering Familiarity and Relatedness

**DOI:** 10.3390/ani11041026

**Published:** 2021-04-05

**Authors:** Robin Walb, Lorenzo von Fersen, Theo Meijer, Kurt Hammerschmidt

**Affiliations:** 1Department of Wildlife Management, University of Applied Sciences Van Hall-Larenstein, Agora 1, 8934 CJ Leeuwarden, The Netherlands; t.h.m.meijer@home.nl; 2Cognitive Ethology Laboratory, German Primate Center, Kellnerweg 4, 37077 Göttingen, Germany; KHammerschmidt@dpz.eu; 3Zoo Nuremberg, Am Tiergarten 30, 90480 Nuremberg, Germany; lorenzo@vonfersen.org

**Keywords:** tapir, *Tapirus indicus*, vocal communication, individual vocalizations, familiarity, relatedness

## Abstract

**Simple Summary:**

Studies in animal communication have shown that many species have individual distinct calls. These individual distinct vocalizations can play an important role in animal communication because they can carry important information about the age, sex, personality, or social role of the signaler. Although we have good knowledge regarding the importance of individual vocalization in social living mammals, it is less clear to what extent solitary living mammals possess individual distinct vocalizations. Malayan tapirs (*Tapirus indicus*) are solitary living forest dwellers that inhabit tropical habitats. We recorded the vocalizations of 14 adult Malayan tapirs (six females and eight males) living in seven European zoos to answer the question of whether Malayan tapirs possess individually distinct vocalizations. Apart from sex-related differences, we found significant differences in the harmonic calls of all subjects. Surprisingly, kinship had no influence on call similarity, whereas familiar animals exhibited significant higher similarity in their harmonic calls compared to unfamiliar or related subjects. The results support the view that solitary animals could have individual distinct calls, like highly social animals. These new insights in the acoustic communication of tapirs provide a solid base to use bioacoustics as conservation tools to protect this endangered species.

**Abstract:**

Studies in animal communication have shown that many species have individual distinct calls. These individual distinct vocalizations can play an important role in animal communication because they can carry important information about the age, sex, personality, or social role of the signaler. Although we have good knowledge regarding the importance of individual vocalization in social living mammals, it is less clear to what extent solitary living mammals possess individual distinct vocalizations. We recorded and analyzed the vocalizations of 14 captive adult Malayan tapirs (*Tapirus indicus*) (six females and eight males) to answer this question. We investigated whether familiarity or relatedness had an influence on call similarity. In addition to sex-related differences, we found significant differences between all subjects, comparable to the individual differences found in highly social living species. Surprisingly, kinship appeared to have no influence on call similarity, whereas familiar subjects exhibited significantly higher similarity in their harmonic calls compared to unfamiliar or related subjects. The results support the view that solitary animals could have individual distinct calls, like highly social animals. Therefore, it is likely that non-social factors, like low visibility, could have an influence on call individuality. The increasing knowledge of their behavior will help to protect this endangered species.

## 1. Introduction

Animals are immersed in a world of signals of smell, sounds, taste, and vision in which they communicate. Communication refers to the way organisms transmit information to each other and it is, to some extent, the medium that holds animal societies together. Reproduction, social status, foraging, and other behavioral aspects are all facilitated by communication [1].

Sound is a widely used channel in animal communication. Individuals of many species sing or call in a repetitive or species-specific way. Vocalizations are, therefore, an important source of information for studies on animal communication [2]. Many species exhibit high inter- and intra-individual variability in their vocal signals (e.g., birds [3], marine mammals [4], primates [5]). Although it is unclear whether these acoustic differences are always meaningful for animals, we have evidence that these signals have the potential to carry different kinds of information, e.g., internal state and external events (predator and food).

In combination with the caller identity, these signals could convey additional information, like the age, sex, personality, or social status. Thus, individual distinct vocalizations can play an important role in animal communication, reducing the uncertainty of the external world (for discussion, see [6]). The individual distinctiveness can range from specific designed signature calls (e.g., [7,8]) to more or less subtle differences in call types, which are sufficient for reliable individual recognition (e.g., [9,10,11]).

Although we have good knowledge of the role of individual vocalization in social living mammals, it is less clear to what extent solitary living mammals show individual differences in their vocalizations. In terrestrial mammals, a part of the individual differences of vocal signals are reflected by physical characteristics. Spectral features are determined by vocal folds and the filter function of the vocal tract (reviewed in [12,13]). For example, the fundamental frequency is the primary determinant of perceived pitch and is controlled by vocal fold size and tension, with longer, thicker, and more relaxed folds producing lower pitched sounds. This makes it very likely that less social living mammals also have individual distinct vocalizations. In addition, familiarity and relatedness could have an influence on call structure [14,15].

To address these questions, we started a study on the vocal communication in zoo-living Malayan tapirs (*Tapirus indicus* Desmarest, 1819). The acoustic communication of Malayan tapirs is almost unknown and there are only few scattered studies. Hunsaker and Hahn [16] studied the vocal communication and their function of five lowland tapirs (*Tapirus terrestris*) at San Diego Zoo. They unveiled that the tapirs uttered two distinct squealing calls and presumed that these contained identical characteristics. A recent study on five captive Malayan tapirs observed four different whistling-type sounds (whistle, whine, squeal 1, and squeal 2) as well as a burp and a hiccup sound [17].

The habitat of tapirs consists of dense primary and secondary tropical rain forest with close access to water [18,19,20]. Due to the restricted visibility in such an environment and the primarily crepuscular and nocturnal activity of tapirs [21], we assumed that the acoustic channel is important for their social communication [22]. Tapirs were originally described as solitary animals; however, different studies revealed that this species could also occur in associations of two or three individuals, forming mating pairs or mother–offspring groups [21,22,23,24].

Spatial studies showed that males have larger home ranges, which overlap with the smaller home ranges of several females, which implies that tapirs likely live in a polygyny mating system [25,26,27,28]. Considering vocal communication, tapirs might use calls to perceive and locate each other [29]. According to Medici [21], frequent and intense vocalization are uttered during courtship, approximation, and mounting.

Malayan tapirs are increasingly threatened due to anthropogenic pressure. Habitat destruction and fragmentation caused by deforestation for agriculture isolates tapir populations and leads to increasing numbers of road kills [30]. Hunting and disease transmission of domestic livestock amplifies this trend [31]. Research to understand the acoustic communication of Malayan tapirs has the additional advantage to gain valuable knowledge regarding the communication of this endangered species [32,33]. The confirmation of individual distinct acoustic signals would enable vocal tagging and allow conclusions about the number and physical characteristics of the signal producers [32,34].

Our specific aims were to describe the vocal repertoire of zoo-living Malayan tapirs and to estimate the degree of individual distinctiveness of their harmonic calls. In addition, we tested whether familiar or related subjects had a higher similarity in their harmonic calls compared with unfamiliar or unrelated subjects. We conducted studies in seven European zoos to extend the number of subjects and to address the question to which extent familiarity and genetic relatedness had an influence on their vocal structure.

## 2. Materials and Methods

### 2.1. Ethics

This study comprises only observational data of zoo animals. The study subjects were not manipulated, and their daily routine was not changed in the context of this study. The authors obtained the permission of the participating zoos to record data of the subjects and also the approval and recommendation from the EAZA Ex-Situ Program (EEP) Coordinator Dr. H. Mägdefrau.

### 2.2. Subjects and Locations

The study was conducted on 14 adult Malayan tapirs (six females and eight males), ranging from the age of 2 to 21 years as well as one young male at the age of two months (Table 1).

### 2.3. Data Collection

Observations took place in seven different European zoos from July until October 2018. Throughout the study, the animals remained in their usual daily husbandry routine. Altogether, 141 h of observation were recorded, 22 h at Zoo Nuremberg, 13 h at Wilhelma Stuttgart, 23 h at Zoo Leipzig, 25 h at Zoo Prague, 28 h at Zooparc de Beauval, 15 h at Zoo Dortmund, and 15 h at Zoo Zlin.

Audio recordings were made during observation hours using a Tascam DR-100 recorder (TEAC Europe GmbH, Germany) and a Sennheiser directional microphone (K6 power module, ME66 recording head (Sennheiser, Germany) covered by a Rycote softie windscreen, in PCM format with a sampling frequency of 48 kHz and a 16-bit amplitude. Due to husbandry reasons individuals living in the same zoo were recorded at the same time. The identity of the caller and the distance between the observer and calling animal as well as whether the call was uttered inside or outside was noted for each call.

### 2.4. Acoustic Analysis

The vocalizations were inspected visually using Avisoft-SASLab Pro (R. Specht Berlin, Germany, version 5.1.20). Based on the acoustic structure, we distinguished three categories: Calls that had a fundamental frequency and several harmonic frequency bands (harmonic calls), noisy calls that were mostly short low-pitched sounds (non-harmonic calls), and a combination of both. For the detailed acoustic analysis, we used only the harmonic calls because they were the most common vocalization (75%), and their structural characteristics had the highest potential for individual differences (Figure 1).

For a detailed structural description, we selected only harmonic calls with a sufficient signal-to-noise ratio undisturbed by other sounds. To obtain an appropriate range to estimate the acoustic features, we reduced the sampling frequency from 48 to 24 kHz and calculated a 1024 pt fast Fourier transformation (FFT), resulting in a frequency resolution of 23 Hz and a time resolution of 5.3 ms. We used the interactive harmonic cursor tool of the custom software program LMA 2018 developed by K. Hammerschmidt [35] to extract 24 acoustic parameters (Table 2).

In total, we had 5885 calls to describe the vocal repertoire and 826 harmonic calls with sufficient quality for detailed acoustic analysis (Appendix A).

### 2.5. Statistics

To calculate the call frequency, we took the ratio of the total recording time and the total number of all calls per individual (calls/recorded hour).

To test for sex and individual differences, we conducted two stepwise discriminant function analyses (DFA) on 826 calls, produced by six females (N = 542) and eight males (N = 284). The stepwise procedure removes highly correlating variables taking care of collinearity. The selection criterion for an acoustic parameter to be entered was *p* = 0.05 and was *p* = 0.1 to be removed from the analysis.

In addition, we cross-validated the classification results with the leaving-one-out method, which involves leaving out each of the cases in turn, calculating the functions based on the remaining n-1 cases, and then classifying the left-out case. We used the Mann–Whitney U-Test to test for sex-related differences in the single acoustic parameter found by the DFA as important parameters to distinguish between male and female calls.

As DFA classification is sensitive to unbalanced samples and likely to overestimate classification results, we ran nested permuted discriminant function analyses [36] on a subset of the data using the same variables to check whether the classification of calls to single subjects was significantly better than chance. For the permutation DFA, we used a function written by Roger Mundry in R [37].

To test the statistical relationship between the acoustic structure, genetic relatedness, and familiarity, we used the F-values of pairwise distances of the stepwise DFA. Four categories were established to differentiate the variety of familiarity and genetic relatedness between subject dyads. Adult tapirs that lived in the same zoo exhibit during the study period were assigned as “familiar” (seven pairs, Table 1). All of those pairs lived together for at least four months.

On the basis of the studbook numbers of the study subjects and their parents, genetically related individuals were identified and categorized as “related”. For this category, only related individuals who were not familiar with each other were considered. Consequently, one dyad of halfsiblings (Pinola–Laila) and two dyads of siblings (Ketiga–Baru and Indah–Nadira) were taken into account. Genetically related subjects that lived in the same zoo for at least four months together in the past were classified as “related & familiar”. Four parent–child relationships within the study population were considered for this category (Laila–Baru, Laila–Ketiga, Copaish–Baru, and Copaish–Ketiga). If neither familiarity nor relatedness could be determined, the dyad was allocated to “not related/not familiar”. The F-values of the stepwise DFA integrate the distance of all acoustic parameters used in the DFA and are frequently applied to investigate the link between the acoustic structure of vocalizations and the genetic relatedness, familiarity, or geographic distance of subjects or populations [15,38,39].

To test for significant relation with relatedness and familiarity, we conducted a linear mixed model based on the F values of 91 pairs, with the four categories of relatedness and familiarity as a fixed factor and subject ID as a random factor (IBM 25). The DFA and mixed model analyses were performed using IBM SPSS 25.

## 3. Results

### 3.1. Vocal Repertoire

During 147 h of recording time, we captured 5885 vocalizations. Based on the sound and visual structure of spectrograms, we were able to classify these vocalizations into 11 call types (Figure 1) over three different categories: non-harmonic calls, harmonic calls, and combined calls.

Overall, the six female tapirs uttered 3645 calls (in 63.86 recording hours) belonging to 10 different call types. Seventy-five (2%) of these were non-harmonic calls, 3537 (97%) were harmonic calls, and 33 (1%) were combined calls. The eight male adult tapirs produced 2195 calls (in 73.19 recording hours) belonging to nine distinct call types, and 1342 (61%) of these were non-harmonic calls, 717 (33%) were harmonic calls, and 136 (6%) were combined calls.

All call types that were recorded for males were also registered for females. Call type “I” was emitted by only one female. The male offspring (Balu) generated 45 calls in total. Thirty-five of them were assigned to call type “J”, which was not produced by any other individual. Therefore, it might be possible that this call type is infant specific, but this requires further investigations.

We distinguished three non-harmonic-call types, which were low-frequency noisy monotonous sounds. Call type “A” was the most frequent call type in males: 58% of all male calls belonged to this type (N = 1274). In females, this call type had a share of 1.5% (N = 56). Except for one female (Pinola), “A” was produced in all individuals even in the two-month-old male. Due to its sound and spectral appearance of two isolated faint sounds, “A” was also termed as “hiccup”. In the case of tapirs who uttered only the “up”, the last part of “A”, this sound was denoted as “B”. This call type was recorded 42 times. The third non-harmonic-call was “C”, which occurred 46 times and arose if tapirs blew air through the mouth or nostrils.

Harmonic calls are clear tonal sounds composed of a fundamental frequency and several harmonics, which are multiple integer frequencies of the fundamental [40]. The spectrographic structure of harmonic calls had a wide variety; hence, seven harmonic call types were distinguished. Call type “D” was the most frequent in this study and was produced by all individuals. However, more than half were emitted by one female (Solo).

Call type “E” was recorded in 13 tapirs and was mainly produced by Pinola. The third most common call type was call type “F”, a high-pitched whistle, generated by two-thirds of all tapirs. Call type “G” was unique due to its high-frequency harmonics running as two consecutive arcs. It was mainly emitted by four females but was also produced by the male Jinak. “H” was the least occurring call type among adult tapirs. Call type “I” was only recorded in the dam Nadira.

Combined calls (“K”) are composed of a harmonic part at the beginning and a non-harmonic part at the ending. We could record 170 combined calls, mostly uttered by males (Table 3).

The usage of the identified call types varied in each subject. Summarized, on average, all females produced 57.1 calls/h, while males uttered 30.0 calls/h (Table 3).

### 3.2. Sex Differences in Harmonic Calls

To determine the degree to which males and females differ in acoustic structure (Table 2), we conducted a stepwise discrimination analysis (DFA) on 826 harmonic calls (Appendix A) that had a sufficient acoustic quality. The DFA required 11 of 24 acoustic parameters (Table 2) to achieve a correct classification of 81.7% (cross-validation = 81.2%, chance-level = 50%). Thus, 435 of 542 (80.3%) female calls and 240 of 284 (84.5%) male calls were correctly classified.

The statistic comparison of the 11 acoustic parameters revealed a significant difference only in call duration. This was due to the more frequent production of longer vocalizations in females. All frequency parameters showed no significant differences between males and females, although F0 min tended to be significantly distinct between sexes (*p* = 0.059; Table 4).

### 3.3. Individual Differences in Harmonic Calls

For individual differences, we tested the 14 adult tapirs. The stepwise DFA used 14 out of 24 acoustic parameters (Table 2) to assign the harmonic calls to the respective subjects (correct assignment = 70.6%, cross validated = 67.1%, and chance level = 7.1%). The individual classification of single subjects ranged from 45.8% to 95.4%, which showed that even the worst classification result was above the chance level. The pDFA on 154 selected calls from 14 subjects indicated that the result was significantly different from chance (*p* < 0.001), confirming that the high individuality in the harmonic calls of tapirs was independent from sex differences or other factors.

### 3.4. Acoustic Similarity in Relation to Relatedness and Familiarity

Based on the F-values of 91 pairs, we conducted a linear mixed model with relatedness and familiarity as two fixed factors and subject ID as a random factor. We found a significant relation with familiarity (F = 7.79, *p* = 0.006) and no relation with relatedness (F = 0.06, *p* = 0.809; Figure 2). Acoustic similarity is expressed by low F-values.

## 4. Discussion

The acoustic analysis of tapir vocalization showed significant sex and individual differences in their harmonic calls, suggesting that individual distinct vocalizations were not restricted to animals living in larger groups or complex societies. The comparison of similarity scores suggests that familiar animals had a more similar vocal structure, whereas relatedness appeared to have no influence on call similarity.

To explain the evolution of vocal distinct vocalization, two major hypotheses were proposed. The distance communication hypothesis expected that calls given over a long distance would be more distinct among individuals than those given at a close distance, because no other cues could be used to enable caller recognition. Support for this hypothesis comes from studies on chimpanzees (*Pan troglodytes*) [41] and mouse lemurs (*Microcebus murinus*) [42].

In both studies, long-distance calls were more individually distinct than calls exchanged over a close distance. The social context hypothesis expected that calls used in social interaction at a close distance should be more distinct than loud calls used to a more general audience. There are several studies that found support for this hypothesis, e.g., red-capped mangabeys (*Cercocebus torquatus*) [43] and female Campbell’s monkeys (*Cercopithecus campbelli campbelli*) [44]. In both species, calls emitted during affiliative social interactions were more individually distinct than their long distance or alarm calls.

However, a study on the vocal repertoire of western gorillas (*Gorilla gorilla*) found high individual distinctiveness in their call types but failed to explain a possible adaptive function by one of the two hypotheses, the distance communication or social communication hypothesis [11], suggesting that both hypotheses could not exclusively explain the evolution of individual distinctive calls.

As in the in case of western gorillas, the individual distinctiveness in tapir harmonic calls cannot be clearly explained by one of the two hypotheses. These calls are not long-distance calls nor are they used in close affiliative interaction. A hypothesis from Rendall and colleagues [45,46] suggests that individual distinctiveness could have evolved as a result of emerging idiosyncrasies and differences in vocal tract development. Maybe this could explain why non-social animals could evolve similar individual differences in their calls as highly social animals.

However, tapir habitats have low visibility, and tapirs are mainly active during the night. Ecological factors, like restricted visual contact, could be the driving force of their individual distinctiveness of calls in a similar way to social communication or communication over longer distances. The fact that we could not detect similarity in call structure with the degree of relatedness supports this hypothesis, because the lack of similarity excludes the possibility that emerging idiosyncrasies in vocal tract development are responsible for individual distinctiveness.

The analysis of tapir harmonic calls suggests that familiarity but not relatedness reflected acoustic similarity between dyads. The effect of familiarity on call structure has been described before. For instance, Snowdon and Elowson [47] reported that pygmy marmosets (*Cebuella pygmaea*), a small new world monkey, modified their call structure when paired with a new partner. Similarly, free-living Campbell’s monkeys shared a higher vocal similarity among closely bonded subjects [48].

In addition, research showed that familiarity can lead to similar vocal structures at the group level in a study on wild chimpanzees [49] and on a study on different groups of Barbary macaques (*Macaca sylvanus*) [50]. Vocal accommodation or auditory facilitation appears to be a likely mechanism to explain the effect of familiarity because nonhuman primates have only limited control over their vocal production and are not able to produce sounds outside their species-specific vocal repertoire [51,52].

Although individual differences in call structure as well as individual recognition is a widespread phenomenon in mammals, the coaction of familiarity and relatedness is well studied only in primates (see the recent review [53]). Marine mammals could use a different method to address individuality as some of them are vocal learners able to learn and produce new sounds. Bottlenose dolphins (*Tursiops truncatus*) produce individual distinctive signature whistles to broadcast their identity. Animals use a whistle from their environment, modify them, and invent a new signal. Bottlenose dolphins can also copy the signature of other dolphins and use these whistles to address the owner. There is also evidence that some other dolphins use signature whistles. Killer whales (*Orcinus orca*) also have whistles, but they do not use them as social signals. They seem to use burst-pulsed sounds for social interactions [54]. These call types are shared within pods rather than being specific to single individuals, although they have individually distinctive features, like the voice cues found in other mammals [55].

In contrast to familiarity, we could not detect significant similarities in relation to relatedness. One explanation could be that the acoustic analysis failed to find the acoustic features reflecting relatedness. However, the high classification results for individuality and sex make it unlikely that the acoustic analysis missed crucial acoustic features. Further, it is possible that phenotypic aspects in call structure are masked by vocal communication.

At least for a certain time in life, related animals grew up by the mother or both parents. In this way, related animals are always to a certain degree in contact with each other, which could make it difficult to separate the two factors in finding animals that are related but unfamiliar with each other. The few studies that were able to include related unfamiliar individuals found mixed results.

One study on mandrills (*Mandrillus sphinx*) showed that familiarity did not impair phenotype matching [56], whereas another study on rhesus macaques (*Macaca mulatta*) failed to find phenotype matching, although the study comprised a high number of subjects (N = 67) [15]. In our study, we had a relatively low number of related but unfamiliar dyads. Therefore, it is necessary to handle the result with caution. More subjects are necessary to come to a final conclusion regarding relatedness and call similarities.

The vocal repertoire of Malayan tapirs described by Naundrup [17] comprised four harmonic and two non-harmonic call types collected from five tapirs. These call types were also found in our study. In addition, we found a further hiccup sound, call type “B”. We also recorded three new harmonic call types, from which one was only uttered by a dam (call type “I”) and one by her juvenile (call type “J”). Both call types might be maternal or infant specific. The combination of harmonic and non-harmonic parts was already mentioned in Naundrup [17] but not defined as separate call types in the vocal repertoire. During our observations, we were able to record this call combination on a regular basis from nearly all subjects, which justifies considering this call combination as its own call type.

The dense forested habitat and the primary nocturnal activity favor the use of acoustic signals not only for larger distances. Under such conditions, visual communication is limited already for short distances. In addition to their acoustic abilities, tapirs have well-developed olfactory communication, including a vomeronasal organ to detect pheromones and scent marks [21,57].

These new insights provide a fundamental base for the use of bioacoustics as a conservation tool in the future for this species. Considering the precarious status of Malayan tapirs, which are classified as endangered by the IUCN (International Union for Conservation of Nature) [19], acoustic surveys can be a useful approach in the determination of population trends particularly on this elusive species in the dense vegetation habitat, which constitute an issue for visual methods [33,58]. The identification of individual-specific acoustic parameters in calls can be used as an alternative tagging technique to physical marks in the studies regarding dispersal distance, site fidelity, survival, and abundance [32].

Further bioacoustic research on this species is necessary, considering certain life circumstances, like birth, mating, flight, or different age classes, to investigate ontogenetic sound usage. This could enable age recognition in acoustic signals as well as the determination of the onset of individual mature calls and vocal sexual dimorphism. Vocal sounds within a broader frequency range should be investigated to determine infrasound usage, which was evidenced in Sumatran rhinos (*Dicerorhinus sumatrensis*) [59]—a close relative of the Malayan tapir.

## 5. Conclusions

Malayan tapirs have a rich vocal repertoire comprising harmonic noisy calls, and call combinations. Their harmonic calls revealed individual characteristics comparable to the individual distinct vocalizations found in many social living animals. This indicates that the emergence of individual distinct vocalization is not solely linked to highly social animals. Tapirs appeared to use their individual distinct vocalizations to communicate with conspecifics.

The study also showed that tapirs that lived together produced calls with more similar acoustic structures compared to non-familiar or related animals. The most likely explanation for familiar animals producing calls with higher similarity is auditory facilitation. This means that an auditory input of a certain call type facilitates the production of the corresponding call type by the listener. This mechanism is independent from vocal learning and could lead to group-specific calls as observed in nonhuman primate species. This mechanism could override possible similarities related to kinship.

These new insights in the acoustic communication of tapirs provide a solid base to use bioacoustics as conservation tools to protect this endangered species.

## Figures and Tables

**Figure 1 animals-11-01026-f001:**
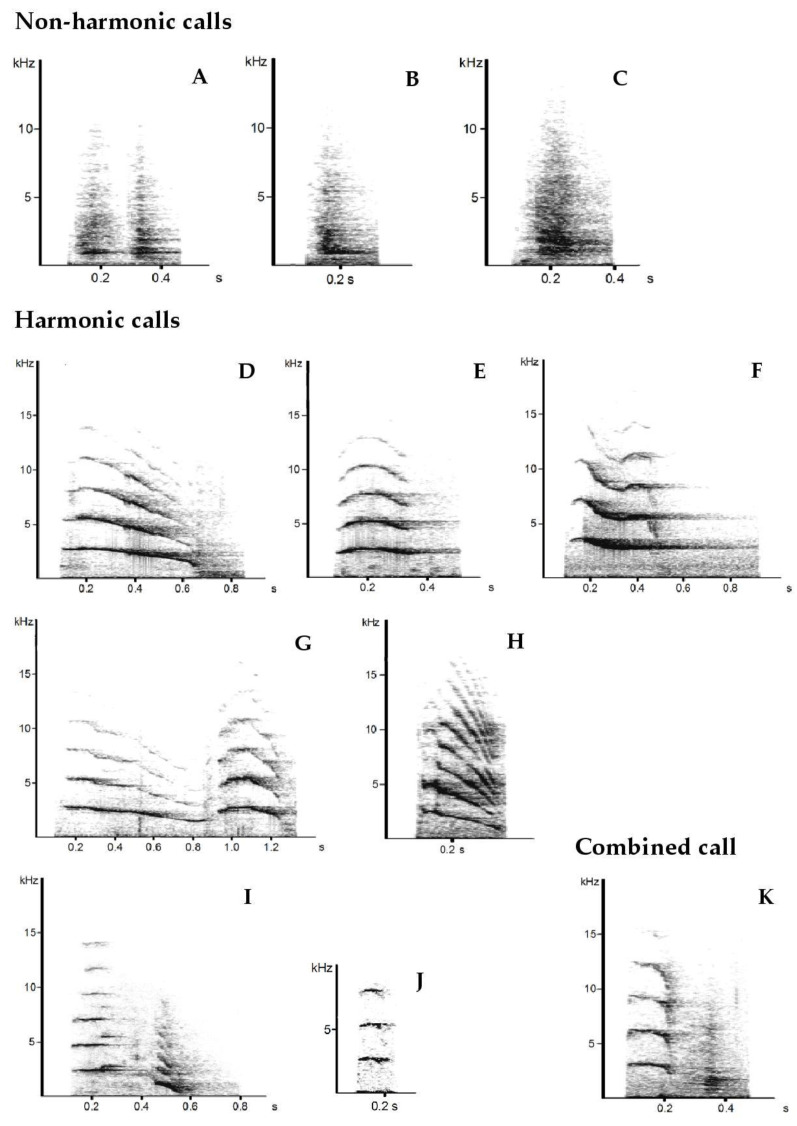
Example spectrograms of the 11 different call types classified in three categories: non-harmonic calls (**A**–**C**), harmonic calls (**D**–**J**), and combined call (**K**).

**Figure 2 animals-11-01026-f002:**
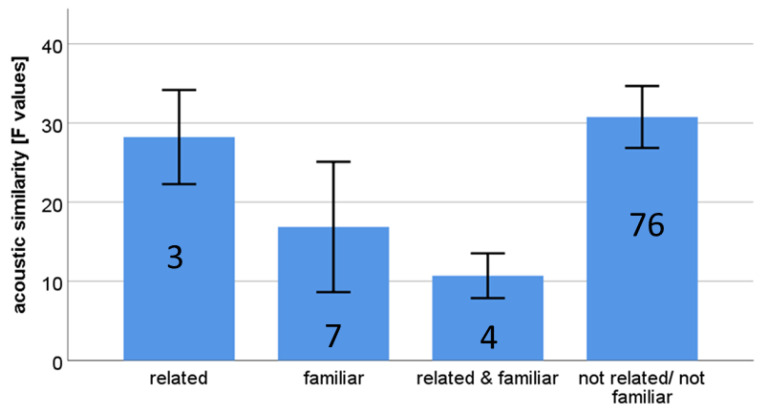
Dyadic acoustic similarity between related, familiar, related and familiar, and dyads that are neither related nor familiar. Acoustic similarity is expressed in F-values (i.e., low F-values refer to high similarity, and high F-values to low similarity). The figure shows the mean ± SEM. Numbers in bars give the number of dyads per category.

**Table 1 animals-11-01026-t001:** The location, demographic data, and close kinship of all study subjects, Stud# is the individual EEP-studbook number (EAZA).

Zoo	Name	Stud#	Sex	Date of Birth	Place of Birth	Sire	Dam
Nuremberg	Pinola	1004	F	28.01.2012	Nuremberg	825	508
	Manado	1003	M	14.01.2013	Lympne	900	655
Leipzig	Laila	942	F	03.04.2009	Nuremberg	501	508
	Copaish	880	M	21.01.2008	Lympne	650	637
Stuttgart	Penang	T6	M	03.08.2016	Rotterdam	816	963
	Ketiga	T2	M	02.06.2016	Leipzig	880	942
Beauval	Harapan	945	F	22.06.2009	Belfast	414	434
	Tioman	919	M	27.12.2008	Amsterdam	466	656
Prague	Indah	939	F	26.09.2008	Edinburgh	786	741
	Niko	498	M	08.12.1996	Berlin	236	321
Dortmund	Solo	T5	F	11.07.2016	Chester	1009	1006
	Jinak	783	M	01.01.2000	Dortmund	416	325
Zlin	Nadira	969	F	03.10.2011	Edinburgh	786	741
	Baru	1013	M	09.02.2013	Leipzig	880	942
	Balu	T13	M	27.07.2018	Zlin	1013	969

**Table 2 animals-11-01026-t002:** Description of the 24 acoustic parameters measured in harmonic calls.

Parameter	Definition
Duration [ms]	Time between begin and end of call
F0 start [Hz]	Fundamental frequency in first time segment
F0 end [Hz]	Fundamental frequency in final time segment
F0 max [Hz]	The maximum fundamental frequency across all time segments
F0 min [Hz]	The minimum fundamental frequency across all time segments
F0 mean [Hz]	The mean fundamental frequency across all time segments
lm mean [Hz] ^1^	Mean frequency difference between original and floating average curve
lm max [Hz] ^1^	Maximum frequency difference between original and floating average curve
F0 loc ^2^	Maximum location of fundamental frequency
F0 trfak ^3^	Factor of linear trend of fundamental frequency
F0 trmean [Hz] ^3^	Mean deviation between F0 and linear trend
F0 trmax [Hz] ^3^	Maximum deviation between F0 and linear trend
Pf start [Hz]	Peak frequency in first time segment
Pf end [Hz]	Peak frequency in final time segment
Pf max [Hz]	The maximum peak frequency across all time segments
Pf min [Hz]	The minimum peak frequency across all time segments
Pf mean [Hz]	The mean peak frequency across all time segments
Pf maxamp [Hz]	Peak frequency of the total maximum amplitude
Pf minamp [Hz]	Peak frequency of the total minimum amplitude
Pf maxloc ^2^	Maximum location of peak frequency
Pf minloc ^2^	Minimum location of peak frequency
Pf maxdif [Hz]	Maximum difference between successive Pf values
Noise mean	Mean ratio of distributed energy measured in Wiener entropy
Noise max	Maximum ratio of distributed energy measured in Wiener entropy

^1^ lm = local modulation. ^2^ calculated as factor of [(1/duration) × location of parameter]. ^3^ tr = global modulation

**Table 3 animals-11-01026-t003:** Distribution of the 11 different call types as well as hours of recording, calls per recording hour, and number of analyzed calls for each individual. To show the frequency in call usage over time, the total number of calls per subject was divided through the total recording time of each subject.

Individual		Non-Harmonic			Harmonic							Combined	Calls Total	Recording Hours	Calls/Hour	Calls Analyzed
Name	Sex	A	B	C	D	E	F	G	H	I	J	K				
Pinola	F	-	1	4	485	74	11	12	-	-	-	19	606	9	64	142
Laila	F	3	1	4	2	3	84	-	-	-	-	-	97	12	8	52
Harapan	F	1	-	1	81	3	-	-	-	-	-	-	86	5	19	25
Indah	F	14	2	4	69	5	2	12	5	-	-	2	115	20	6	53
Solo	F	8	-	-	1848	3	4	222	1	-	-	12	2098	8	273	174
Nadira	F	30	1	1	373	1	-	7	2	228	-	-	643	10	64	96
**Total**	F	56	5	14	2858	89	101	253	8	228	-	33	3645	64	57	542
Manado	M	640	29	17	16	1	27	-	-	-	-	66	796	9	84	21
Copaish	M	9	1	1	7	18	12	-	-	-	-	2	50	12	4	16
Penang	M	31	3	-	13	6	127	-	-	-	-	12	192	4	45	79
Ketiga	M	46	-	-	33	14	168	-	-	-	-	2	263	5	56	87
Tioman	M	177	1	1	25	-	-	-	-	-	-	30	234	5	51	13
Niko	M	72	2	6	18	7	-	-	13	-	-	6	124	20	6	23
Jinak	M	248	-	2	102	17	17	44	14	-	-	11	455	8	57	34
Baru	M	51	1	4	12	1	5	-	-	-	-	7	81	10	8	11
**Total**	M	1274	37	31	226	64	356	44	27	-	-	136	2195	73	30	284
Balu juv.	M	6	-	1	2	-	-	-	-	-	35	1	45	10	4	-
**Total all ind.**		1336	42	46	3086	153	457	297	35	228	35	170	5885	147	40	826

**Table 4 animals-11-01026-t004:** Differences in the acoustic parameters used by the discriminant function analysis (DFA) between female and male calls using the Mann–Whitney U-test. Due to the no significant differences in all frequency parameters, we made no correction for multiple testing.

Parameter		Median Female	Median Male	U	*p*	r
Noise max		0.56	0.52	19.000	0.573	0.17
F0 min	[kHz]	1.51	1.95	9.000	0.059	0.52
Duration	[s]	0.26	0.14	3.000	0.005	0.72
Noise mean		0.36	0.33	20.000	0.662	0.14
Pf maxdif	[kHz]	3.42	3.55	22.000	0.852	0.07
F0 trmean	[kHz]	0.19	0.15	19.000	0.573	0.17
F0 maxloc	[kHz]	0.00	0.00	13.000	0.181	0.38
F0 end	[kHz]	1.76	2.41	12.000	0.142	0.41
F0 max	[kHz]	3.02	3.14	23.000	0.950	0.03
F0 trfak		-0.21	-0.20	22.000	0.852	0.07
Pf min	[kHz]	0.92	1.28	14.000	0.228	0.35

## Data Availability

All data needed to evaluate the conclusions in the paper are present in the paper. Additional data related to this paper may be requested from the corresponding author.

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
