# Peer review of "Individual Differences in the Vocal Communication of Malayan Tapirs (Tapirus indicus) Considering Familiarity and Relatedness"

_animals, 2021, doi:10.3390/ani11041026_

Round 1
Reviewer 1 Report
Dear Authors,
attached please find a pdf with comments.
Regards

Author Response
Dear reviewer,
Thank you for your positive assessment and your valuable suggestion. We appreciate your time and effort providing feedback on our manuscript.
Please find attached our revised manuscript entitled “Individual differences in the vocal communication of Malayan tapirs (Tapirus indicus) considering familiarity and relatedness” by Robin Walb, Lorenzo von Fersen, Theo Meijer, and Kurt Hammerschmidt.
Changes in the manuscript are highlighted (color-coded).
Regarding your comments:
1: Rewrite the title of this article because it sounds like a conclusion.
We thank you for pointing this out. We have changed the title (see above).
2: Please introduce authority for this subspecies scientific name.
We appreciate your suggestion and included the authority of the study species at the first mention in the introduction. However, the reference of every authority of every species that is mentioned in the manuscript does not fit the style of the journal.
3: Major revision necessary-the same information as in simple summary.
We made sure that the content of the simple summary coincide with the content of the abstract.
4: The results do not allow obtaining clear conclusions due to the small number of individuals analyzed.
We agree and have revised the phrasing of abstract and simple summary.
5: In the abstract, you state that the study was conducted on 14 Malayan tapirs.
We recorded 15 Malayan tapirs in total, but one tapir was just two month old. We decided to exclude the young animal from the acoustic analysis because possible individual differences could be also due to age. Recordings of the young animal were only used to visualize the vocal repertoire of tapirs. We apologize if this was confusing. We make this now clearer (methods 2.2).
6: There are 9 males listed in the table?
Nine male individuals are listed including the young animal of two months. The vocalizations of eight adult male Malayan tapirs were used in the acoustic analysis.
7: Please improve resolution of Fig. 1.
We replaced figure 1 with a figure with higher resolution. Thank you for pointing this out.
To make sure grammar and spelling errors are not in the article, we submitted the manuscript to the English editing service of Animals.
Reviewer 2 Report
Overall, I found this to be an interesting and well designed experiment documenting differences in vocalisations between zoo-housed Malayan tapirs. The methods are well described and the topic of vocalisation has not been extensively studied. This study therefore identifies several novel aspects of tapir communication, including new calls and similarities in calls as a result of proximity rather than relatedness.
The work is well formatted and generally, there are only a few errors, most of which relate simply to word choice or grammar. I have highlighted these where relevant in the PDF file.
Other than this, the only other key area to consider is the application to industry: do we know what each of the calls is used for, and the circumstances when they were used by the tapirs? This may be beyond the scope of the current study but is certainly a point to highlight for future study.

Author Response
Dear reviewer,
Thank you for your overall positive assessment and your valuable suggestion. We appreciate your time and effort providing feedback on our manuscript.
Please find attached our revised manuscript entitled “Individual differences in the vocal communication of Malayan tapirs (Tapirus indicus) considering familiarity and relatedness” by Robin Walb, Lorenzo von Fersen, Theo Meijer, and Kurt Hammerschmidt.
Changes in the manuscript are highlighted (color-coded).
Due to a suggestion of one reviewer we changed the title (see above).
1: Spelling errors, confused words:
We made all changes accordingly to your suggestions. Thank you for your careful review. To make sure grammar and spelling errors are not in the article, we submitted the manuscript to the English editing service of Animals.
2: Different numbers of animals in abstract and method (2.2):
We have a different number of animal in abstract and methods because we recorded 15 tapirs in total, but one tapir was just two month old. We decided to exclude the young animal from the acoustic analysis because possible individual differences could be also due to age. We make this now clearer (methods 2.2).
3: A lot of the background here relates to primates which are generally very social. A bit more information on tapir communication (what the different calls are used for) would be useful here.
To our knowledge there is no study on call usage or call meaning. Although we noticed the context of call usage in our study and did some preliminary playback experiments we a far from the point to give some scientific answers. The playbacks showed that tapir’s response to whistles with increased vigilance and approach behavior, which indicates that these calls seem to have some importance for their social communication.
4: Errors and writing of scientific names in references:
We thank you for pointing this out. We shortened journal names and changed the writing of scientific names in italic style.
Reviewer 3 Report
This manuscript is well written and the experimental design is appropriate. I'm not sure the application of the findings is as profound to the conservation of the species as stated. The paper, however, is good in its present state and needs only minor language adjustments.
Line 91. “…as cohesion manner…” does not make sense, please revise.
Line 315. “…can hardly explained…” is missing a word. The same line “… these calls are no long distance calls…”
Author Response
Dear reviewer,
Thank you for your overall positive assessment and your suggestions for improvement. We appreciate your time and effort providing feedback on our manuscript.
Please find attached our revised manuscript entitled “Individual differences in the vocal communication of Malayan tapirs (Tapirus indicus) considering familiarity and relatedness” by Robin Walb, Lorenzo von Fersen, Theo Meijer, and Kurt Hammerschmidt.
Changes in the manuscript are highlighted (color-coded).
Thank you for pointing out language errors. We changed the mentioned parts. To make sure grammar and spelling errors are not in the article, we let perform an English editing service.
Due to a suggestion of one reviewer we changed the title (see above). In addition we made some minor changes in the manuscript due to the comments of the editor and two other reviewers (see highlighted text).
Round 2
Reviewer 1 Report
The Authors have made the suggested improvements. I have no additional comments.
Regards